# Medium- and Long-Term Electric Vehicle Ownership Forecasting for Urban Residents

**Zhao-Xia Xiao [1,]*, Jiang-Wei Jia [1,]*, Xiang-Yu Liu [1], Hong-Kun Bai [2], Qiu-Yan Li [2] and Yuan-Peng Hua [2]**

1. Tianjin Key Laboratory of Intelligent Control of Electrical Equipment, Tiangong University, Tianjin 300387, China; liuxiangyu@tiangong.edu.cn
2. State Grid Henan Economic Research Institute, Zhengzhou 450052, China; baihongkun@sohu.com (H.-K.B.); hnlxy2003@163.com (Q.-Y.L.); 13783661352@163.com (Y.-P.H.)
* Correspondence: xiaozhaoxia@tiangong.edu.cn (Z.-X.X.); 2130070802@tiangong.edu.cn (J.-W.J.)

**Abstract:** With the rapid development of electric vehicles (EVs) in Chinese cities, accurately forecasting the number of EVs used by urban residents in the next five years and more long term is beneficial for the government to adjust industrial policies of EVs, guide the rational planning of urban charging facilities and supporting distribution network, and achieve the rational and orderly development of the EV industry. The paper considers the advantages of using the grey GM(1,1) prediction model to predict the short-term ownership of EVs by urban residents. Then, by forecasting the number of EV users in a certain city in the future and predicting the number of private vehicles in the future, the boundary conditions for long-term year ownership of EVs by residents are determined. Combined with historical data and short-term forecast data generated by the grey prediction model, the model parameters that include the innovation coefficient and imitation coefficient of the Bass model are trained using a genetic algorithm. Finally, the Bass model is used for medium- to long-term ownership forecasting from 2023 to 2040. The prediction error for the target year is provided. The simulation results indicate that the ownership of resident EVs in this city will experience rapid growth in the next five years.

**Keywords:** ownership forecasting; EV; grey GM(1,1) prediction model; Bass model; prediction error

## 1. Introduction

With the challenges of global warming, serious environmental pollution, and depletion of traditional fossil energy, countries are actively promoting the development of renewable energy. As an environmentally friendly means of transportation, EVs have significant advantages such as low cost and zero pollution and are being widely promoted and adopted [1–3]. By the end of 2023, the number of new energy vehicles in China reached 20.41 million, of which pure EVs reached 15.52 million, accounting for the main part [4]. In addition, the rapid development of EVs not only helps balance the peak and valley load of the power grid but also promotes the economic development of the upstream and downstream of the EV industry chain and increases the employment rate, which has a positive role in promoting [5]. Therefore, a convenient and accurate forecast of EV ownership is of great practical significance for the orderly development of EVs, the rational layout of charging facilities, and the planning of urban distribution network.

The forecast of EV ownership mainly adopts the methods based on the time series model, regression model, and diffusion model [6]. Li et al. [7] established a prediction model of EV ownership based on comprehensive prediction, applied three prediction models of grey prediction, back propagation (BP) neural network, and long-term memory (LSTM) network to predict EV ownership, obtained the prediction results of the single prediction model, and used the entropy weight method to assign weights to the prediction results of the single prediction model. Finally, it is pointed out that the comprehensive prediction model can provide higher prediction accuracy than the single prediction model,

but the model is sensitive to data distribution, needs to determine the weight range, requires high data quality, has high computational complexity, and lacks theoretical support. Li et al. [8] used the long short-term memory networks (LSTM) model and the system dynamics (SD) model, respectively, to predict EV ownerships. Based on the errors of these two models, the LSTM-SD combined prediction model for EV ownership is proposed, which improves the accuracy of prediction results. However, although the (LSTM-SD) model improves some prediction accuracy, it is not conducive to popularization due to its complex construction, high data requirements, high computing resource requirements, and difficult parameter optimization. Messianic et al. [9–11] predicted the number of EVs by analyzing the influence of imitation coefficient, innovation coefficient, and market potential on prediction accuracy in the Bass model. Although the Bass model provides a simple and intuitive method to analyze and predict market performance and can accurately predict the growth trend of product sales, the model does not consider the complexity and individual differences of consumer behavior, thus affecting the accuracy of the model's prediction. Guo et al. [12–14] proposed an improved Bass model to predict EV ownership. Compared with the traditional Bass model, the improved Bass model adjusted the relevant parameters in the model, making the predicted EV ownership more consistent with the actual situation. Wang et al. [15–17] analyzed that, due to the short popularization time of EVs, there are less data, while GM(1,1) grey prediction model is suitable for a small amount of data and has good prediction accuracy in the short term. Therefore, the GM(1,1) grey prediction model is adopted to make short-term prediction of EV ownership. Wang et al. [18] elastic coefficient method and thousand-people ownership method were used to establish a prediction model and forecast EV ownership in Shanxi Province, China. Due to the short time for the large-scale application of EVs, the predicted results of EV ownership differ greatly from the actual results. Therefore, the error of EV ownership predicted by the elastic coefficient method and thousand-people ownership method is large.

In the research on the forecast of EV ownership, many scholars believe that the grey prediction model has good forecast accuracy in the short term and the Bass model has the advantage of comprehensively considering internal and external influencing factors. Under the condition of reasonable model parameters, it has good forecast accuracy for medium- and long-term EV ownership. Therefore, this paper considers the advantages of the grey GM(1,1) prediction model to forecast EV ownership in the short term, combines the historical EV ownership data with the grey model short-term forecast data, and uses Bass model parameters to accurately forecast EV ownership in cities in the recent five years and long-term years.

## 2. EV Ownership Forecasting Framework

Aiming at the three main problems of insufficient historical data on EV ownership, inaccurate forecasting trend and growth rate of EV ownership, and low forecasting accuracy of target year and long-term year ownership, this paper first makes use of the advantages of the grey GM(1,1) prediction model, such as the small sample required, the model's data, and the high short-term forecast accuracy, to forecast the EV ownership in large communities in the past three years and uses the forecast data to expand the historical data of EV ownership. Then, by forecasting the growth of the population between 20 and 60 years old in the city, forecasting the number of fuel vehicles and the proportion of new energy vehicles that fuel vehicles withdraw from the market, we determine the long-term year and forecast the number of EVs in the long-term year more accurately. Finally, combining the historical data and the data forecasted by the grey prediction model, the genetic algorithm is used to train the innovation coefficient and imitation coefficient of the Bass model first, and the Bass model is used to forecast the EV ownership from the target year to the prospective year. The forecasting framework of EV ownership is shown in Figure 1.

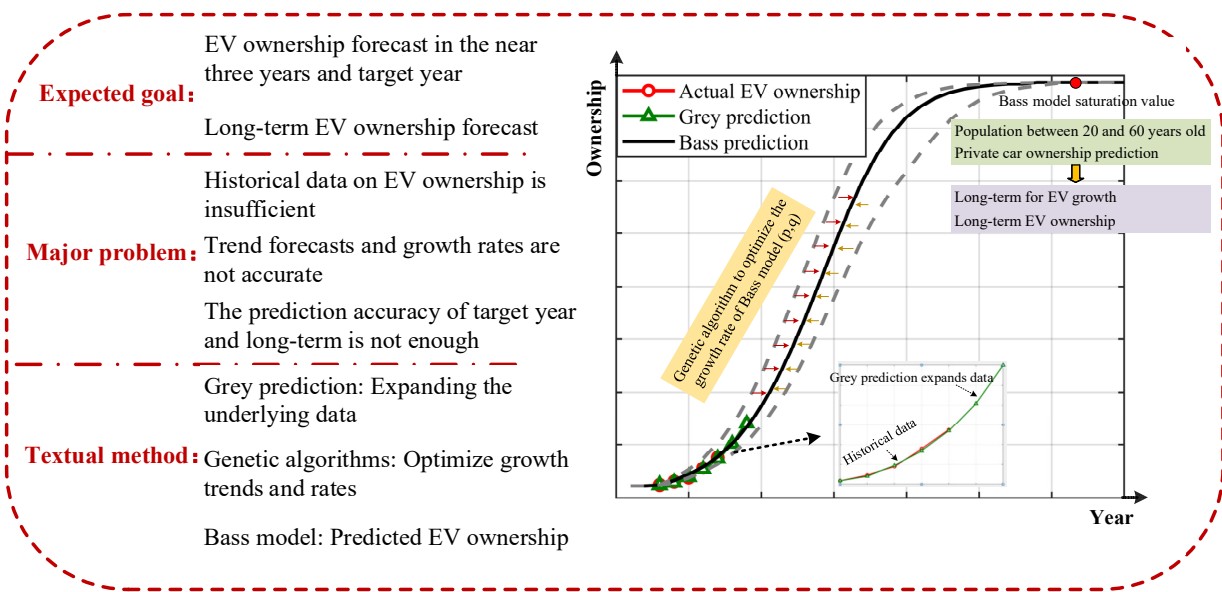

**Figure 1.** EV ownership forecast block diagram.

### 3. Short-Term EV Ownership Forecast Based on Grey GM(1,1) Model

The research object of grey system theory is the small data uncertainty system with incomplete data or poor information. The theory mainly extracts valuable information by mining part of the known information to correctly describe and effectively monitor the operation behavior and evolution law of the system. The grey prediction model is a method used to describe the development trend of things and make short-term forecasts, and its forecast accuracy is relatively high. However, when making a long-term forecast, the model does not take into account factors such as internal development law and external changes of things, so the data of a long-term forecast are susceptible to other factors, which easily lead to distortion [19].

Since there is little historical data on urban EV ownership, the functional relationship between the data is unknown, and urban EV ownership is only forecasted in the short term, the grey prediction model is selected for forecast. In the short term, the forecast difference between the GM(1,1) model and GM(1,N) model in the grey prediction model is very small [20], so this paper uses the grey GM(1,1) prediction model to make a short-term forecast of EV ownership, and the specific thinking framework is shown in Figure 2.

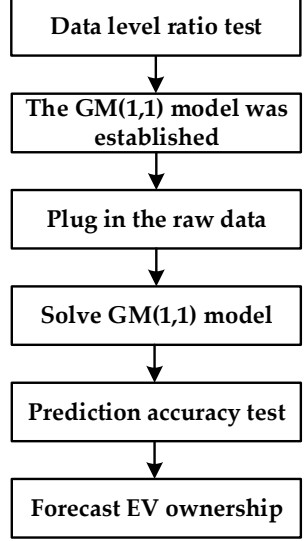

**Figure 2.** The GM(1,1) model forecasts EV ownership.

First of all, the historical data of urban EVs are tested by level ratio and calculation parameter $\lambda_k$. If $\lambda_k$ is in the interval $\left(e^{-\frac{2}{n+1}}, e^{\frac{2}{n+2}}\right)$, it means that the GM(1,1) model can be used; otherwise, it is necessary to add the passable grade-ratio test constant c to the annual holding data, and then the holding forecast result needs to subtract this constant [21].

$$\lambda_k = \frac{x^{(0)}(k-1)}{x^{(0)}(k)} \quad k = 2, 3, \cdots, n \tag{1}$$

where $x^{(0)}(k)$ is the historical ownership data of EVs in year $k$ and $n$ is the historical number of ownership data in years.

Secondly, the grey GM(1,1) prediction model is established by constructing a cumulative generation sequence.

The cumulatively generated sequence is:

$$x^{(1)}(k) = \sum_{k=1}^{n} x^{(0)}(k) \quad k = 2, 3, \cdots, n \tag{2}$$

The grey GM(1,1) predict model is:

$$x^{(0)}(k) + ax^{(1)}(k) = b \tag{3}$$

where $a$ is the development coefficient, reflecting the speed of data development; $b$ is the gray action, reflecting the degree of influence of influencing factors on the development trend.

To solve the GM(1,1) model, it is necessary to construct a first-order differential equation that accumulates the generated sequence and year and solve it by the least square method. The differential equation is:

$$\frac{dx^{(1)}}{dt} + ax^{(1)} = b \tag{4}$$

The time response sequence is obtained as follows:

$$\hat{x}^{(1)}(k) = \left(x^{(0)}(1) - \frac{\hat{b}}{\hat{a}}\right)e^{-\hat{a}(k-1)} + \frac{\hat{b}}{\hat{a}} \quad k = 2, 3, \cdots, n \tag{5}$$

where $\hat{a}$ is the development coefficient estimated by the least square method; $\hat{b}$ is the grey action estimated by the least square method.

The forecasted value of EV ownership is:

$$\hat{x}^{(0)}(k) = \hat{x}^{(1)}(k) - \hat{x}^{(1)}(k-1) \quad k = 2, 3, \cdots, n \tag{6}$$

where $\hat{x}^{(0)}(k)$ is the forecasted value of EV ownership by the GM(1,1) model.

Finally, the forecast accuracy of the GM(1,1) model is tested by residual test and relative error test.

Residual test:
$$\varepsilon^{(0)}(k) = x^{(0)}(k) - \hat{x}^{(0)}(k) \quad k = 2, 3, \cdots, n \tag{7}$$

Relative error test:

$$e(k) = \varepsilon^{(0)}(k)/x^{(0)}(k) \quad k = 2, 3, \cdots, n \tag{8}$$

The relative error test value is compared to Table 1 and the accuracy level must meet the second level or above to pass the test.

**Table 1.** Relative error test grade reference.

| Accuracy Level | Level 1 | Level 2 | Level 3 | Level 4 |
|---|---|---|---|---|
| Mean relative error | 0.01 | 0.05 | 0.10 | 0.20 |

## 4. Long-Term EV Ownership Forecast

Long-term EV ownership is determined by forecasting the proportion of long-term EV users and the proportion of long-term EV ownership in private vehicle ownership.

From the perspective of EV users, the 30–50-year-old group is the main group of new energy vehicle consumption, accounting for more than 60%. For people between 20 and 60 years old, new energy vehicles account for 99%, and the number of people between 20 and 60 years old can be set as the upper limit of EV ownership in the future year. According to the statistical yearbook data of the selected target city statistics bureau, the historical population data of the city between 20 and 60 years old and the proportion of the population are found and the long-term forecast is made.

From the perspective of the forecast of the proportion of EV ownership in private vehicle ownership in the future year, the proportion of EV ownership in private vehicle ownership in the future year is taken as the upper limit of EV ownership. According to the "Research on the Withdrawal Schedule of China's Traditional Fuel Vehicles" released by the Energy and Transportation Innovation Center, China's fuel vehicles will withdraw from the market in 2040 at the earliest. It is generally believed that China will not be able to replace fuel vehicles with new energy vehicles until 2050, as shown in Table 2. Internationally, the current proposed time frame for the ban is between 2025 and 2040. Private fuel vehicles in Tier I and II cities will gradually withdraw from the market in 2030, and Tier III and IV cities are expected to withdraw from the market in 2035 and 2040, respectively, and EVs are expected to account for 75% of private vehicles in 2040 [22].

**Table 2.** Traditional fuel vehicles exit regional hierarchy and representative regions.

| Hierarchy | The Main Basis and Representative Region |
|---|---|
| I | • large cities (such as Beijing, Shanghai, etc.);<br>• Functional demonstration areas (such as Hainan, Xiongan, etc.). |
| II | • First cities of traditional automobile purchase restriction (such as Tianjin, Hangzhou, etc.);<br>• Provincial capitals of key regions (such as Shijiazhuang, Taiyuan, Zhengzhou, Jinan, Xi'an, etc.);<br>• Leading cities of new-energy vehicle promotion, core cities of industrial clusters, and coastal cities with economic development (such as Chongqing, Qingdao, Chengdu, Changsha, etc.). |
| III | • Key areas of the Blue Sky Defense War, such as North China (Hebei, Henan, and Shandong), Yangtze River Delta (Jiangsu, Zhejiang, and Anhui), and Fen-Wei Plain region (Shanxi);<br>• New energy automobile industry cluster regions, such as the pan-Pearl River Delta (Guangzhou, Fujian), Central (Hunan, Hubei, Jiangxi);<br>• Other new energy vehicle promotion or low-carbon development demonstration cities such as Guiyang. |

## 5. Medium- and Long-Term Vehicle Ownership Forecast of EVs Based on the Bass Model

Bass model, as a diffusion model, is a model that dynamically evolves the market size of new products from a macro perspective. It regards the diffusion process of new products as the process of potential groups gradually transforming into consumer groups and has the advantage of comprehensively considering internal and external factors [23,24]. When the EV market is in its infancy, EVs can be considered as a new durable product and introduced into the automotive market through penetration strategies. The Bass model can feedback well on the development and diffusion process of new products, and the characteristics of the model are in line with the current development status of EVs [25]. However, the model

has high requirements on the original historical data. If the original data are insufficient, the fitting effect of the model will be poor, which will affect the subsequent forecast results.

The Bass model in the discrete-time domain can be expressed as [26]:

$$f(t) = Mp[1 - \frac{F(t)}{M}] + qF(t)[1 - \frac{F(t)}{M}] \tag{9}$$

where $F(t)$ is the total amount of new products accumulated up to time $t$; $f(t)$ refers to the number of new products added during the $t$ period; $M$ is the maximum market potential, indicating the number of new durable products that grow to saturation; $p$ is the innovation coefficient, indicating the degree of influence of external media publicity on the diffusion of emerging durable products; and $q$ is the imitation coefficient, indicating the degree of influence of internal word-of-mouth communication on the spread of emerging durable products.

In this paper, the medium- and long-term forecast of urban EV ownership is made based on the Bass model, in which the maximum value $M$ of the model is determined according to the number of EV users in the future year or the proportion of EVs in private vehicles in the future year. Combining the historical data of urban EV ownership and the data forecasted by the grey prediction model, the innovation coefficient $p$ and imitation coefficient $q$ are determined by the historical data of genetic algorithm training. The parameter flow of the genetic algorithm training model is shown in Figure 3. Finally, the number of EVs in the city from the base year to the prospect year is forecasted year by year. The specific idea framework is shown in Figure 4.

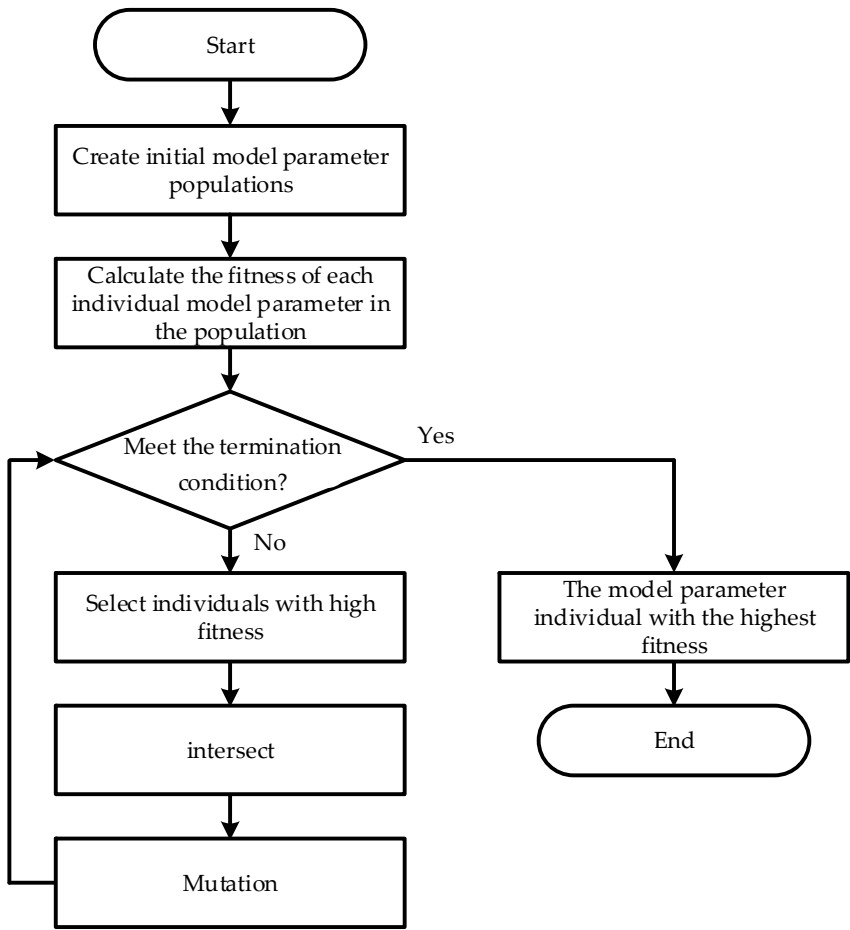

**Figure 3.** Parameter flow chart of genetic algorithm training model.

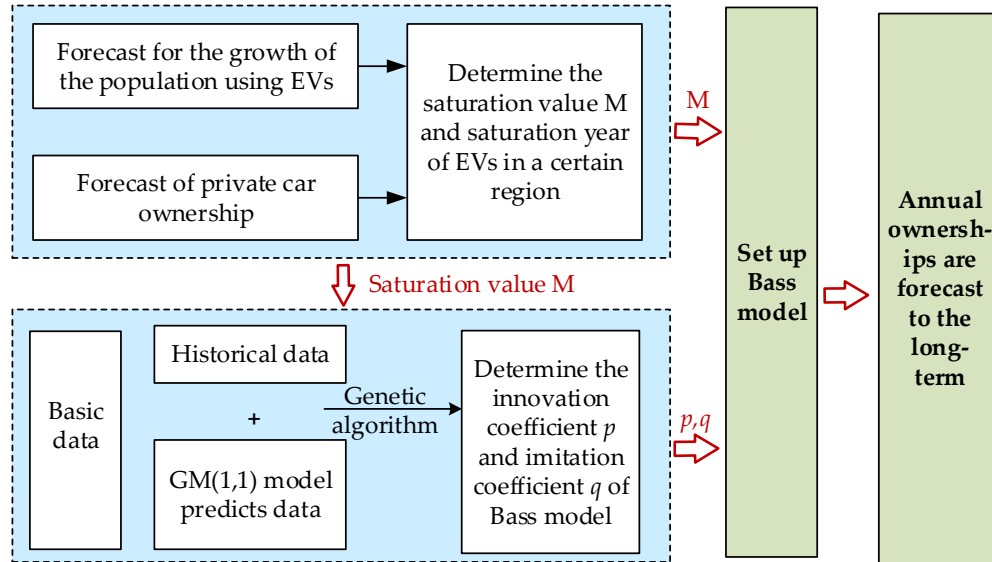

**Figure 4.** Bass model forecasting EV ownership.

## 6. Example Analysis

Taking Tianjin City of China as an example, the medium- and long-term predicted model of urban EV ownership constructed in this paper is analyzed. The population of the city in 2022 was 13.63 million and the number of private vehicles was 2.242 million [27]. Therefore, it is calculated that the number of private vehicles in the city is 165. At the same time, the city's EV ownership in 2022 was 375,000, accounting for 16.72%.

### 6.1. Grey Prediction Model Forecast of EV Ownership

With 2022 as the base year and the next fifth year as the target year (2027), the grey prediction model is combined to forecast the ownership of EVs in the next two years. The historical data on EV ownership are shown in Table 3.

**Table 3.** Historical year EV ownership data.

| Year | 2018 | 2019 | 2020 | 2021 | 2022 |
|---|---|---|---|---|---|
| Ownership/vehicle | 117,600 | 147,000 | 189,000 | 278,000 | 374,700 |

Combined with the grey prediction model, EV ownership in Tianjin from 2018 to 2024 is predicted and the prediction results are shown in Figure 5. In the short term, the grey prediction model has a good forecasting effect of EV ownership.

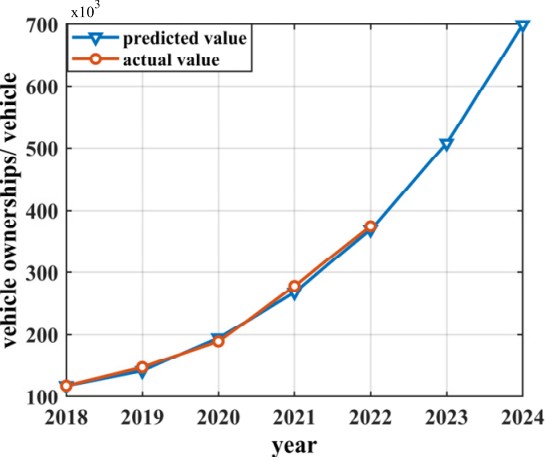

**Figure 5.** Grey prediction model forecasts EV ownership.

As can be seen from Tables 1 and 4, the average relative error is 0.029, and the average relative error test method is used to test it. When the accuracy level of the relative error reaches the first or second level, the forecast can be made, and the origin error is 0 less than 0.02, it can be concluded that the research forecast results meet the accuracy requirements. From the forecast results, the city's EVs will continue to develop rapidly in the next two years, reaching 507,900 and 699,300 in 2023 and 2024, respectively.

**Table 4.** Comparison between actual data and forecasted data of the grey prediction model.

| Year | The Number of EVs/Vehicle | Forecast Data | Residual Error | Relative Error/% |
|------|---------------------------|---------------|----------------|------------------|
| 2018 | 117,600 | 117,600 | 0 | 0 |
| 2019 | 147,000 | 141,400 | 0.56 | 3.83 |
| 2020 | 189,000 | 194,600 | −0.56 | −2.97 |
| 2021 | 278,000 | 267,900 | 1.00 | 3.61 |
| 2022 | 374,700 | 368,900 | 0.58 | 1.54 |

### 6.2. The Largest Market Potential for EVs

The ownership and growth rate of private vehicles in Tianjin are shown in Table 5. Based on the 6-year historical data, the fact that the population between 20 and 60 years old will not grow rapidly after 2025, and the "carbon peaking and carbon neutrality" target of carbon emission reaching its peak in 2030, Tianjin, as a new first-tier city, will reach its peak carbon emission faster than other cities in China and set 2025 as the turning point for the growth rate of private vehicles. The growth rate curve of private vehicle ownership in this city is fitted by using the fitting tool in MATLAB, as shown in Figure 6.

**Table 5.** Private vehicle ownership and growth rate in target cities.

| Year | 2017 | 2018 | 2019 | 2020 | 2021 | 2022 |
|------|------|------|------|------|------|------|
| Forecast vehicle ownership/vehicle | 1,703,600 | 1,738,800 | 1,768,900 | 1,885,800 | 2,061,600 | 2,242,300 |
| Private vehicle growth rate/% | 2.90 | 2.07 | 2.07 | 6.61 | 9.32 | 8.77 |

Based on the data in Table 5, the function of the fitting curve of the growth rate of private vehicle ownership in the city can be expressed as:

$$f_{growth}(t) = ae^{-\left(\frac{t-b}{c}\right)^2} \tag{10}$$

where *a*, *b*, and *c* are the fitting curve parameters, *a* = 0.114, *b* = 2025, and *c* = 6.12.

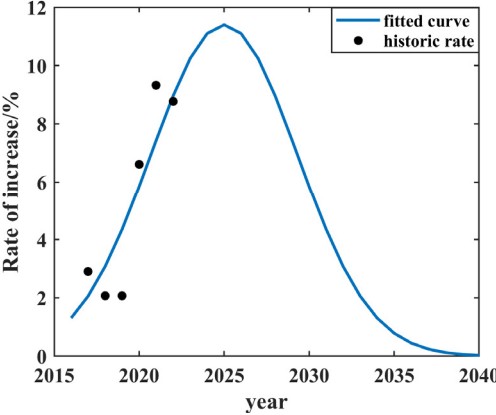

**Figure 6.** Fitting curve of private vehicle ownership growth rate.

As can be seen from Figure 6, if the target city's private vehicle ownership in 2022 is used as the base value to forecast private vehicle ownership, urban private vehicle ownership will grow rapidly in the next five years, and the growth rate of private vehicle ownership will approach 0 in 2040 and private vehicle ownership will reach saturation. Therefore, 2040 is set as the vision year for the forecast of urban private vehicle ownership. The growth rate of private vehicle ownership is calculated according to Equation (10), and the forecasted results of private vehicle ownership are shown in Table 6.

**Table 6.** Forecast of private vehicle ownership in Tianjin.

| Year | Forecast Vehicle Ownership |
| --- | --- |
| 2023 | 2,472,000 |
| 2024 | 2,746,400 |
| 2025 | 3,055,000 |
| 2030 | 4,642,900 |
| 2035 | 5,204,400 |
| 2040 | 5,251,600 |

According to this paper's method for determining the maximum market potential of EVs in the future year and the forecast data in Table 6, since Tianjin is a Tier II city, it is expected that EVs will account for 75% of private vehicles in 2040, that is, 3,938,700 vehicles.

According to the growth trend of the permanent resident population from 2001 to 2021 in the Statistical yearbook of the Tianjin Bureau of Statistics, it is found that it reached the saturation value around 2012 and then fluctuated around 14 million, as shown in Figure 7. Therefore, according to the analysis, the permanent resident population in the prospective year is about 14 million.

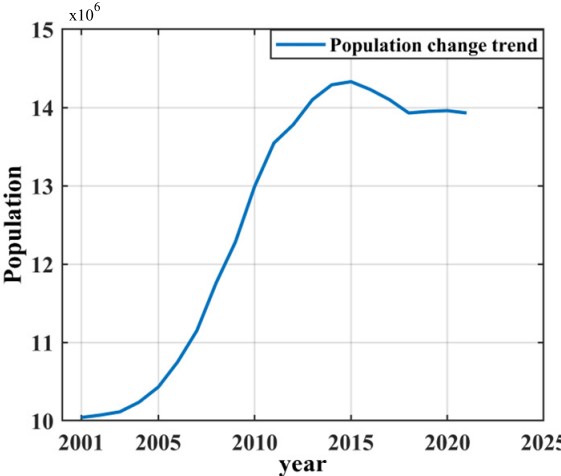

**Figure 7.** Population change trend of the city from 2001 to 2021.

Based on the statistical yearbook of Tianjin City, the basic data of the population aged 20–60 from 2017 to 2021 are obtained in Table 7, and the growth rate of the population aged 20–60 from 2022 to 2030 is fitted with the Gaussian function in the Matlab toolkit, as shown in Figure 8.

**Table 7.** Basic data of the population aged 20-60, 2017-2021.

| Year | 2017 | 2018 | 2019 | 2020 | 2021 |
| --- | --- | --- | --- | --- | --- |
| Population | 5,526,400 | 5,626,200 | 5,666,900 | 5,729,200 | 5,778,900 |

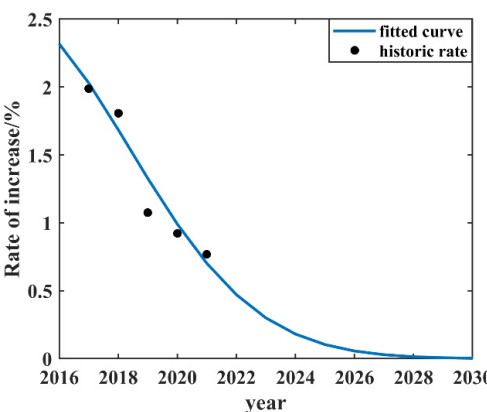

**Figure 8.** Fitting curve of population growth rate between 20 and 60 years old.

Based on the data in Table 7, the function of the fitting curve of the city's population growth rate between 20 and 60 years old is consistent with Formula (11), where $a$ = 2.5776, $b$ = 2014, and $c$ = 6.1365.

Combined with the growth rate fitting curve, the population growth rate of cities tends to be zero in 2030, so it can be considered that the population of this age group will not change significantly from 2030 to 2040. Based on the population growth forecast from 2022 to 2030, the population aged 20 to 60 in 2040 will be about 5,846,500, as shown in Figure 9.

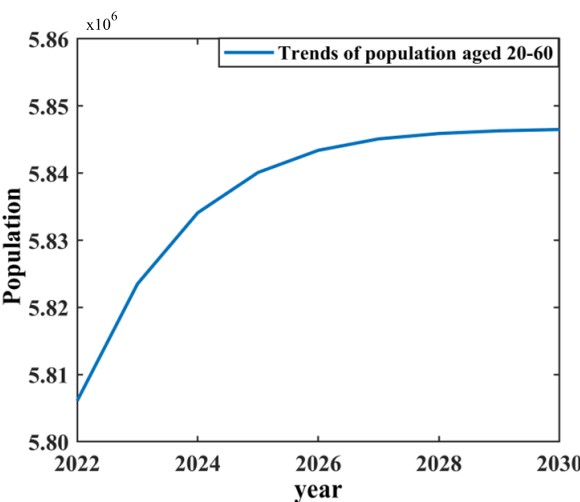

**Figure 9.** Demographic trends in the 20–60 age group, 2022–2030.

Considering that 75% of the population aged 20–60 or private vehicle ownership in the future year is considered as the upper boundary condition of EV ownership, this paper chooses a smaller number group as the maximum capacity of EVs.

### 6.3. Bass Model Forecasts EV Ownership

By compiling a genetic algorithm to train $p$ and $q$ influence coefficients in the Bass model, given the relatively limited historical data of EV ownership and the accuracy of the grey prediction model in the short-term forecast, this paper combines historical data of EV ownership with short-term forecast data of the grey prediction model to train innovation coefficient $p$ and imitation coefficient $q$. Among them, the population value is 100, the number of iterations is 1000, the crossover probability is 0.7, the mutation probability is 0.001, and the parameter values in the model are obtained, as shown in Table 8.

**Table 8.** Bass model parameter data.

| Model Parameter | Maximum Market Potential/Vehicle | Innovation Coefficient $p$ | Coefficient of Imitation $q$ |
|---|---|---|---|
| Numerical value | 3,938,700 | 0.0006 | 0.35 |

After the influence coefficient of the Bass model is obtained by the genetic algorithm, the coefficient is brought into the model to calculate EV ownership. The fitting curve of the Bass model is shown in Figure 10, and the comparison between the predicted data and the actual data per thousand people is shown in Table 9.

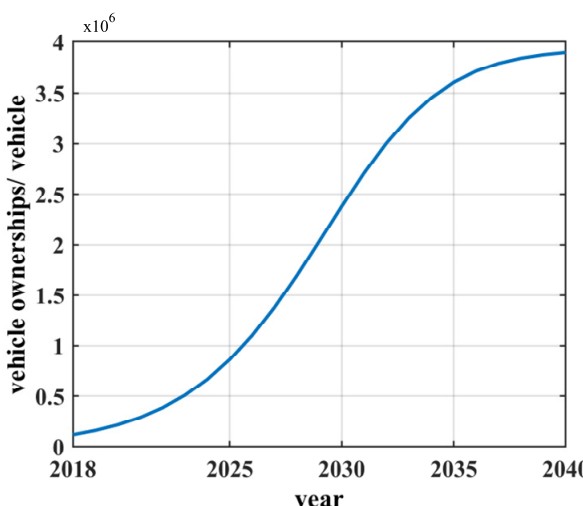

**Figure 10.** Bass model fits the number of EVs.

**Table 9.** Comparison between actual data and forecasted data of the Bass model.

| Year | EV Ownership/ Thousand | Forecast Data/Thousand | Residual Error | Relative Error/% |
|---|---|---|---|---|
| 2018 | 9 | 9 | 0 | 0 |
| 2019 | 11 | 12 | −1.28 | −8.72 |
| 2020 | 14 | 16 | −2.68 | −14.16 |
| 2021 | 20 | 21 | −1.14 | −4.09 |
| 2022 | 27 | 28 | −1.07 | −2.86 |
| 2023 | 37 | 37 | −0.13 | −0.26 |
| 2024 | 51 | 48 | 3.28 | 4.69 |

As can be seen from Figure 10 and Table 9, the Bass model has a good performance in forecasting the ownership of EVs. Compared with the real data, it is found that the residual difference between the forecasted data and the real data is small each year. It leads to a large relative error. The Bass model was used to forecast the EV ownership in Tianjin from 2023 to 2040 and the iterative process results of EV ownership per 1000 people in the city were obtained, as shown in Table 10.

**Table 10.** Iterative process of urban EV per 1000 people ownership.

| Year | Private Vehicle Ownership/Thousand | EV Ownership/Thousand |
|---|---|---|
| 2023 | 179 | 37 |
| 2024 | 199 | 48 |
| 2025 | 222 | 62 |
| 2027 | 272 | 100 |
| 2035 | 377 | 261 |
| 2040 | 381 | 282 |

## 7. Conclusions

This paper forecasts the long-term ownership of EVs by urban residents. Firstly, a grey prediction model is used to forecast the number of EVs owned by residents in the city for the next two years. Secondly, by forecasting the number of residents using EVs in the future, the number of private vehicles in the future, and the proportion of EVs in the private vehicles in the future, the number of EVs in the future can be determined. Then, using the historical data of urban residents using EVs and the data predicted by the grey prediction model, the innovation coefficient p and imitation coefficient q in the Bass model are determined through genetic algorithm training. They use the Bass model to forecast the long-term ownership of EVs by urban residents.

This paper has two main contributions. Firstly, it proposes a method for predicting the boundary of EV ownership among urban residents. Secondly, in response to the insufficient historical data on the number of EVs, a proposed method of supplementing historical data is provided.

The forecasting of the number of EVs owned by urban residents provides a scientific basis for the rational layout of charging facilities and the planning and construction of urban distribution networks. In practical development, the dynamic changes in the number of EVs are related to policies, urban planning, road construction, and economic growth. Consideration of these factors is a further research goal.

**Author Contributions:** Conceptualization, Z.-X.X.; methodology, J.-W.J.; software, J.-W.J. and X.-Y.L.; validation, Z.-X.X. and J.-W.J.; data curation, H.-K.B., Q.-Y.L. and Y.-P.H.; writing—original draft preparation, J.-W.J. and Z.-X.X. All authors have read and agreed to the published version of the manuscript.

**Funding:** This work was supported by the National Key Research and Development Program of China (2023YFE0198100).

**Data Availability Statement:** The dataset used in this article can be obtained from the corresponding author under reasonable request.

**Conflicts of Interest:** The authors declare no conflict of interest.

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
