# Peer review of "Medium- and Long-Term Electric Vehicle Ownership Forecasting for Urban Residents"

_wevj, doi:10.3390/wevj15050212_

Round 1

Reviewer 1 Report

Comments and Suggestions for Authors

To adapt the industrial policy for electric vehicles, to guide the rational planning of urban charging points and the supporting distribution network, and to achieve the rational and orderly development of the electric vehicle industry, requires forecasting the number of cars. The article carries out such analyzes for Chinese cities. The paper considers the advantages of using the Grey GM(1,1) prediction model to predict the short-term ownership of EVs by urban residents.

1.      The authors should explain what they mean in formula (9) and in what units they are expressed: "M - maximum market potential", "p - innovation coefficient; "q - imitation coefficient".

2.      On the basis of what data was the curve in Fig. 6 describing the growth of private vehicles in the city selected?

3.      Is this motel universal for Chinese cities? Can it be easily transferred to the forecast for European countries? How does the central control policy of the Chinese state influence the decision-making processes regarding the number of cars owned by individual owners?

4.      Can the proposed algorithms take into account the availability of minerals and materials for the production of electric cars, e.g. lithium for the production of batteries?

Author Response

Response to Reviewer 1 Comments

1. Summary

First of all, we would like to thank you, the assistant editor and the reviewers for your insightful and helpful comments. We have carefully read the comments from you and the reviewers, and we have revised our paper according to the comments and suggestions. The changes in the manuscript are marked up using the “Track Changes”.

2. Questions for General Evaluation

Reviewer’s Evaluation

Response and Revisions

Does the introduction provide sufficient background and include all relevant references?

Yes

Are all the cited references relevant to the research?

Yes

Is the research design appropriate?

Yes

Are the methods adequately described?

Can be improved

Are the results clearly presented?

Can be improved

Are the conclusions supported by the results?

Can be improved

3. Point-by-point response to Comments and Suggestions for Authors

Comments 1: The authors should explain what they mean in formula (9) and in what units they are expressed: "M - maximum market potential", "p - innovation coefficient; "q - imitation coefficient".

Response 1:

The meaning of unit in formula (9) is supplemented.” M is the maximum market potential, indicating the number of new durable products that grow to saturation;” “p is the innovation coefficient, indicating the degree of influence of external media publicity on the diffusion of emerging durable products;”” q is the imitation coefficient, indicating the degree of influence of internal word-of-mouth communication on the spread of emerging durable products.” Please refer to the main body of the article for details.

Comments 2: On the basis of what data was the curve in Fig. 6 describing the growth of private vehicles in the city selected?

Response 2:

The curve describing the growth of private cars in the selected cities in Figure 6 is based on the data in Table 5. There are two reasons why the article sets 2025 as the turning point for the growth rate of private cars. Reason 1: The population aged 20-60 in Figure 9 will grow slowly after 2025. Reason 2: China's "double carbon" goal is to reach the peak of carbon emissions in 2030, and Tianjin, as a new first-tier city, will reach the peak of carbon emissions faster than the country. According to the historical growth rate data of private cars and the confirmed inflection point, the growth rate curve of private cars is fitted by Gaussian fitting function (Figure 6).

Comments 3: Is this motel universal for Chinese cities? Can it be easily transferred to the forecast for European countries? How does the central control policy of the Chinese state influence the decision-making processes regarding the number of cars owned by individual owners?

Response 3:

(1) The electric vehicle ownership prediction model presented in this paper can be used as a general model to predict the electric vehicle ownership in other Chinese cities.

(2) The ownership prediction model in this paper can be used to predict the ownership of electric vehicles in European countries, but the maximum market potential M of the model needs to be adjusted according to the specific conditions of European countries. For example: according to the growth rate of private cars to determine the future year; electric vehicle users in the European countries in which age group; private cars reached the saturation value of the population of this age group; according to the relevant policies of European countries, to reach the future year of electric vehicles in the proportion of private cars forecast.

(3) The Chinese government will introduce policies to encourage individuals to buy electric cars. For example: electric vehicle purchase tax exemption policy; Vehicle purchase subsidy policy; Free charging or preferential charging policy; Preferential parking policies and additional license plate quota policies for individuals.

Comments 4: Can the proposed algorithms take into account the availability of minerals and materials for the production of electric cars, e.g. lithium for the production of batteries?

Response 4:

Yes, the available properties of the minerals and materials required for the production of electric vehicles can be considered by the algorithm proposed in the paper. Specific manifestations:

1. Use multi-factor grey prediction model to make short-term prediction of electric vehicle ownership. The model considers factors such as the price of batteries, the number of charging points, and subsidies for electric vehicles. Since the multi-factor grey prediction model is difficult to collect the data of the influencing factors and the prediction accuracy is not much different from that of the single factor prediction model in the short term, the single factor GM(1,1) model is chosen in this paper.

2. Adjust the model parameters in Bass model by considering the factors affecting the ownership of electric vehicles. The paper can take into account factors such as lithium in the production of batteries, but this will greatly increase the difficulty of data collection of the model, which is not conducive to the popularization of the model. This paper considers the historical data of electric vehicle ownership, population data and private car ownership data, which are three easy to find and important influencing factors.

4. Response to Comments on the Quality of English Language

Point 1: I am not qualified to assess the quality of English in this paper

Response 1: The full text English has been further optimized.

5. Additional clarifications

If you are not clear about anything, we welcome your criticism and correction. Finally, thank you for your comments.

Reviewer 2 Report

Comments and Suggestions for Authors

The manuscript presents a forecasting method for medium and long-term electric vehicle ownership, which I find to be an interesting topic. The figures included in the manuscript are of suitable quality. However, I believe that some revisions are necessary before recommending this manuscript:

1- It is advisable to use relevant references for the equations, if necessary, to provide clarity and support for the theoretical aspects of the model.

2- The references section requires several corrections, including the addition of page numbers or paper numbers where applicable, to ensure accuracy and completeness.

3- The authors should incorporate the most recent references into the introduction and elaborate on the differences between their work and previous efforts. The current reference list appears to be outdated.

4- Proper citation of previous works in the introduction section is essential. For instance, phrases such as “The authors in [10]...” should be revised to “Wang et al. [10]...”.

5- I encourage the authors to summarize the advantages and disadvantages of their model compared to other relevant models. This would provide readers with a clearer understanding of the contributions and limitations of the proposed approach.

6- Some figures may require additional discussions to better elucidate their relevance to the study.

Author Response

Response to Reviewer 2 Comments

1. Summary

First of all, we would like to thank you, the assistant editor and the reviewers for your insightful and helpful comments. We have carefully read the comments from you and the reviewers, and we have revised our paper according to the comments and suggestions. The changes in the manuscript are marked up using the “Track Changes”.

2. Questions for General Evaluation

Reviewer’s Evaluation

Response and Revisions

Does the introduction provide sufficient background and include all relevant references?

Must be improved

Are all the cited references relevant to the research?

Must be improved

Is the research design appropriate?

Can be improved

Are the methods adequately described?

Must be improved

Are the results clearly presented?

Must be improved

Are the conclusions supported by the results?

Must be improved

3. Point-by-point response to Comments and Suggestions for Authors

Comments 1: It is advisable to use relevant references for the equations, if necessary, to provide clarity and support for the theoretical aspects of the model.

Response 1:

References to the grey GM(1,1) model and Bass model equations have been added. Specific location in the article of the literature [21] and [26].

Comments 2: The references section requires several corrections, including the addition of page numbers or paper numbers where applicable, to ensure accuracy and completeness.

Response 2:

The whole reference has been modified as required. Please refer to the References section of the article for details.

Comments 3: The authors should incorporate the most recent references into the introduction and elaborate on the differences between their work and previous efforts. The current reference list appears to be outdated.

Response 3:

The most recent references have been incorporated into the introduction as requested. Please refer to the introduction section for details.

Comments 4: Proper citation of previous works in the introduction section is essential. For instance, phrases such as “The authors in [10]...” should be revised to “Wang et al. [10]...”.

Response 4:

The description of the introduction has been revised as requested. Please refer to the introduction section for details.

Comments 5: I encourage the authors to summarize the advantages and disadvantages of their model compared to other relevant models. This would provide readers with a clearer understanding of the contributions and limitations of the proposed approach.

Response 5:

It has been modified as required. With reference to the introduction part of the article, it summarizes the models and research contents of others, and points out the advantages and disadvantages of the models.

Comments 6: Some figures may require additional discussions to better elucidate their relevance to the study.

Response 6:

Figures 6 through 9 are further explained here. In this paper, the maximum market potential M of electric vehicles is determined by the population between 20 and 60 years old in long-term and the proportion of electric vehicles in private cars in long-term.

(1) Figures 7 to 9 mainly show how to determine the number of people aged 20-60 in long-term. FIG. 7 According to the data of permanent population of Tianjin in the past 21 years, it is analyzed that the future permanent population will fluctuate around 14 million. Figure 8 is a fitting curve of population growth rate between 20 and 60 years old by combining population data between 20 and 60 years old and Gaussian algorithm in the past five years. Figure 9 shows the growth rate data fitted according to Figure 8, which predicts the demographic trends of the 20-60 age group in 2022-2030.

(2) Explanation of Figure 6: The curve describing the growth of private cars in the selected cities in Figure 6 is based on the data in Table 5. There are two reasons why the article sets 2025 as the turning point for the growth rate of private cars. Reason 1: The population aged 20-60 in Figure 9 will grow slowly after 2025. Reason 2: China's "double carbon" goal is to reach the peak of carbon emissions in 2030, and Tianjin, as a new first-tier city, will reach the peak of carbon emissions faster than the country. According to the historical growth rate data of private cars and the confirmed inflection point, the growth rate curve of private cars is fitted by Gaussian fitting function.

4. Response to Comments on the Quality of English Language

Point 1: English language fine. No issues detected.

Response 1: The full text English has been further optimized.

5. Additional clarifications

If you are not clear about anything, we welcome your criticism and correction. Finally, thank you for your comments.

Round 2

Reviewer 1 Report

Comments and Suggestions for Authors

All my suggestion has been taken into account and indicated problems have been solved. Additional comments, which have been added on a request of the other reviewers also improve the scientific level of the paper. Therefore, in my opinion, the paper in the current version is suggested to be accepted in WEVJ Journal.

Reviewer 2 Report

Comments and Suggestions for Authors

The authors have appropriately revised their manuscript in accordance with my comments. However, a minor issue remains: when citing other references, authors should only include the family name of the first or corresponding author for proper referencing of other works. Please revise the introduction accordingly.

Li et al. [7] established...

Li et al. [8] used...

Researchers predicted.........[9-11].

and other parts should also be rewritten correctly.